# Gut Microbiota—Adversary or Ally? Its Role and Significance in Colorectal Cancer Pathogenesis, Progression, and Treatment

**DOI:** 10.3390/cancers16122236

**Published:** 2024-06-15

**Authors:** Katarzyna Chawrylak, Magdalena Leśniewska, Katarzyna Mielniczek, Katarzyna Sędłak, Zuzanna Pelc, Timothy M. Pawlik, Wojciech P. Polkowski, Karol Rawicz-Pruszyński

**Affiliations:** 1Department of Surgical Oncology, Medical University of Lublin, Radziwiłłowska 13 St., 20-080 Lublin, Poland; 56646@student.umlub.pl (K.C.); 56506@student.umlub.pl (M.L.); katarzyna.sedlak@umlub.pl (K.S.); zuzanna.pelc@umlub.pl (Z.P.); wojciech.polkowski@umlub.pl (W.P.P.); karol.rawicz-pruszynski@umlub.pl (K.R.-P.); 2Department of Surgery, The Ohio State University Wexner Medical Center and James Comprehensive Cancer Center, Columbus, OH 43210, USA; tim.pawlik@osumc.edu

**Keywords:** gut microbiota, colorectal cancer, dysbiosis, human gut microbiome, probiotics

## Abstract

**Simple Summary:**

In 2022, colorectal cancer (CRC) ranked third globally among diagnosed malignancies. Treatment typically involves a multimodal approach based on surgery and systemic chemotherapy. The human gut microbiota, comprising over 35,000 bacterial species, is heavily influenced by diet, impacting food absorption and inflammation development. Dysbiosis in the gut microbiota is strongly linked to CRC development. Recent findings suggest that the gut microbiome has a role in modulating cancer treatment effectiveness and toxicity. Therapeutic strategies like probiotics show promise in CRC treatment. This review aims to describe the current state of knowledge regarding the involvement of gut microbiota in CRC pathogenesis and its potential therapeutic implications in treating this disease, emphasizing emerging interventions.

**Abstract:**

In 2022, colorectal cancer (CRC) was the third most prevalent malignancy worldwide. The therapeutic approach for CRC typically involves a multimodal regimen. The human gut microbiota comprises over 35,000 bacterial species. The composition of the gut microbiota is influenced by dietary intake, which plays a crucial role in food absorption, nutrient extraction, and the development of low-grade inflammation. Dysbiosis in the gut microbiota is a key driver of inflammation and is strongly associated with CRC development. While the gut microbiome influences CRC initiation and progression, emerging evidence suggests a role for the gut microbiome in modulating the efficacy and toxicity of cancer treatments. Therapeutic strategies targeting the gut microbiome, such as probiotics, hold promise as effective interventions in the modern therapeutical approach to CRC. For example, Microbiota Implementation to Reduce Anastomotic Colorectal Leaks (MIRACLe) implementation has resulted in improvements in clinical outcomes, including reduced incidence of anastomotic leakage (AL), surgical site infections (SSIs), reoperation, as well as shorter recovery times and hospital stays compared with the control group. Therefore, this review aims to describe the current state of knowledge regarding the involvement of the gut microbiota in CRC pathogenesis and its potential therapeutic implications to treat CRC.

## 1. Introduction

In 2022, colorectal cancer (CRC) was the third most prevalent malignancy worldwide, with an estimated incidence of 1,926,425 cases, representing almost 10% of all cancer diagnoses. Additionally, CRC accounted for 904,019 deaths globally, which was the second most common cancer-related reason for mortality thereby constituting 9.3% of all cancer-related deaths [1].

Risk factors associated with CRC include advanced age, male gender, familial predisposition, obesity, sedentary lifestyle, high consumption of red meat, and a medical history that includes abdominopelvic radiation exposure, acromegaly, hereditary hemochromatosis, or prior ureterosigmoidostomy [2].

The therapeutic approach for CRC typically involves a multimodal regimen (Figure 1) comprised of surgery along with preoperative and postoperative chemotherapy (CTH) or radio-chemotherapy (RCT), including intraoperative radiotherapy (IORT) in selected patients [3,4]. Data have underscored the potential benefits of preoperative interventions to mitigate postoperative complications and enhance treatment outcomes. Neoadjuvant CTH (NAC), often employing a combination of Folinic acid (FA), 5-fluorouracil (5-FU), oxaliplatin (OXA), irinotecan (CPT-11), capecitabine, and mitomycin C (MMC), aims to induce tumor regression with the objective of downstaging tumors and thereby improving local disease control [5]. Treatment options for stage 0 (carcinoma in situ) CRC may include local excision or simple polypectomy conducted via endoscopy, or alternatively, resection followed by bowel anastomosis may be considered if the tumor size exceeds the feasibility of local excision [6]. Surgical intervention is indicated for stage I-III patients, which may be conducted via open surgery, laparoscopic-assisted colorectal surgery (LACS), or robot-assisted colorectal surgery (RACS) [7]. Evidence suggests that RACS may yield superior outcomes compared to alternative approaches, as indicated by lower rates of blood loss, complications, mortality, bleeding, and ileus, along with shorter hospital stays, while the incidence of anastomotic leakage (AL) was comparable between RACS and open surgery, and the incidence of wound infection was similar between RACS and laparoscopic surgery [8]. In instances of obstructive colon cancer, upfront surgery might be undertaken, typically in an emergency setting [9]. In the case of rectal cancer patients, the predominant treatment for the stage I disease involves proctectomy or proctocolectomy, utilized in 61% of cases, with approximately half of the individuals also undergoing neoadjuvant radiotherapy (RTH) and/or CTH, while the remainder solely undergo upfront surgery [7]. In cases of metastatic CRC, adjuvant therapy following local resection is recommended. Among the most efficacious regimens is the combination of OXA with 5-FU and leucovorin [10].

The human intestine hosts over 1000 species and a colony of approximately 10^14^ microorganisms. These microbial communities play crucial roles in maintaining physiological homeostasis, including energy metabolism, gut barrier system reinforcement, epithelial cell survival support, and protection against external pathogens. Recent research has elucidated the profound impact of the gut microbiome on host physiology [11]. Gut microbiota in healthy adults consists mainly of the following anaerobe bacteria species: *Bacteroides*, *Eubacterium*, *Bifidobacterium*, *Fusobacterium*, *Peptostreptococcus*, and *Atopobium*, with less coexistence in the population of aerobic bacteria, such as *Enterococci*, *Lactobacilli*, *Enterobacteriaceae*, and *Streptococci* [12].

Dysbiosis, characterized by imbalances in microbial composition, was linked to various diseases, encompassing gastrointestinal and metabolic disorders [13]. Alterations in the gut microbiome can be induced by dietary patterns or shifts in environmental factors, and several studies have highlighted their association with CRC pathogenesis via inflammatory pathways, microbial metabolites, and virulence factors [14,15]. The following changes in intestinal microbiome were found in CRC patients: increased diversity of the *Clostridium leptum* and *C. coccoides* subgroups, *Bacteroides*/*Prevotella*, *Fusobacterium*, presence of such pathogens as *Pseudomonas*, *Helicobacter* and *Acinetobacter*, decreased *Bifidobacterium*, *Faecalibacterium*, *Blautia* butyrate-producing bacteria and *Firmicutes* and *Lactobacillus* [14]. While the gut microbiome influences CRC initiation and progression, emerging evidence suggests a role for the gut microbiome in modulating the efficacy and toxicity of cancer treatments [16].

Therapeutic strategies targeting the gut microbiome, such as probiotics, hold promise as effective interventions in the modern therapeutical approach to CRC. These agents are anticipated to be pivotal in modulating the gut microbiome to enhance CRC treatment outcomes [17].

Therefore, this review aims to describe the current state of knowledge regarding the involvement of the gut microbiota in CRC pathogenesis and the potential therapeutic implications of treating CRC. We aimed to offer a novel perspective on the utilization of gut microbiota in a multimodal treatment approach for CRC, focusing primarily on alleviating post-CTH symptoms and minimizing the risks of intraoperative and postoperative complications.

## 2. Materials and Methods

A comprehensive literature review was conducted utilizing PubMed and Google Scholar databases, employing search terms such as “gut microbiota”, “colorectal cancer”, “dysbiosis”, “human gut microbiome”, “metastatic colorectal cancer” and “probiotics” with articles selected from the years 2015 to 2024, with a few older, frequently cited articles also included. Filters were applied to include clinical trials, meta-analyses, and systematic reviews. Preference was given to articles published within the last decade, with seminal works cited prior to this timeframe also considered. Exclusion criteria involved articles not written in English and book chapters. In order to maintain quality standards, the included studies underwent evaluation concerning their relevance, methodology, and significance. Systematic data extraction was carried out, encompassing key discoveries, study design specifics, participant characteristics (limited to CRC patients across all stages of the disease, various age groups, or animal subjects), interventions applied, resultant outcomes, and whether the study successfully reached any defined endpoints.

## 3. Characteristics of the Human Gut Microbiome

Frank et al. discovered that the human gut microbiota comprises over 35,000 bacterial species [16]. Le Chatelier et al., in their gut microbiome study, identified two distinct groups: high gene count (HGC) and low gene count (LGC), each associated with implications for health and disease [17]. The HGC microbiome is characterized by an abundance of *Butyrivibrio crossotus*, *Anaerotruncus colihominis*, *Fecalibacterium* sp. and *Akkermansia* sp., along with a high *Akkermansia (Verrucomicrobia): Ruminococcus torque*/*gnavus* ratio. Notably, the HGC microbiome exhibits features conducive to digestive health, including increased butyrate production, hydrogen production propensity, the establishment of a methanogenic/acetogenic ecosystem, and reduced hydrogen sulfide production. Individuals with an HGC microbiome demonstrate a more robust gut microbiome function and a lower prevalence of metabolic disorders and obesity. Conversely, LGC individuals harbor a higher proportion of pro-inflammatory bacteria such as *Bacteroides* and *Ruminococcus gnavus*, both associated with inflammatory bowel disease. Other members of LGC bacteria include *Porphyromonas*, *Staphylococcus, Parabacteroides*, *Campylobacter*, *Dialister*, and *Anaerostipes*. Furthermore, key bacterial metabolites in LGC individuals, such as modules for β-glucuronide degradation, aromatic amino acid degradation, and dissimilatory nitrite reduction, are known to exert malicious effects [17]. Functional aspects of the normal gut microbiota are depicted in Figure 2.

## 4. Influence of the Diet on the Gut Microbiome 

The composition of the gut microbiota is influenced by dietary intake, which plays a crucial role in food absorption, nutrient extraction, and the development of low-grade inflammation [18]. Recent research indicates that diets high in animal and saturated fats can induce alterations in the gut microbiota, characterized by increased levels of lipopolysaccharides (LPS), trimethylamine-N-oxide (TMAO), and decreased levels of short-chain fatty acids (SCFA) [19]. The mechanism underlying LPS absorption, a potential trigger for systemic inflammation, remains unclear but may involve enhanced plasma LPS filtration into the lymphatic system during fat absorption. LPS, predominantly found in the outer membrane of Gram-negative bacteria such as *Proteobacteria*, act as endotoxins and are absorbed into intestinal capillaries for transport with chylomicrons [20].

High intake of monounsaturated fatty acids (MUFA) was associated with reduced levels of *Bifidobacteria* spp. and slightly increased levels of *Bacteroides* spp. Conversely, higher omega-6 polyunsaturated fatty acids (PUFA) intake was linked to decreased *Bifidobacteria* spp. counts [21].

Furthermore, studies highlight the impact of high-fat dietary patterns, such as regular consumption of red meat, leading to elevated *Bacteroides* spp. concentrations [22]. Similarly, diets rich in saturated fats, such as milk, were shown to promote delta-proteobacteria growth, notably *Bilophila wadsworthia* [23]. High-fat diets contribute to dysbiosis, resulting in a notable reduction in *Roseburia* spp. [24].

Research indicates that increased consumption of plant-based foods correlates with a higher abundance of *Prevotella* bacteria in the gastrointestinal microbiota [25]. In a comparative study involving pediatric populations from Burkina Faso (14 healthy children) and Italy (15 healthy children), Tomova et al. assessed the impact of dietary patterns on gut microbiota composition. The Italian cohort adhered to a low-fiber diet typical of Western dietary practices, while the Burkina Faso cohort consumed a diet rich in fiber and resistant starch. Analysis revealed that the Burkina Faso group exhibited elevated levels of *Bacteroidetes*, *Prevotella*, and *Xylanibacter taxa* alongside diminished *Firmicutes* populations. Furthermore, the African participants demonstrated heightened production of SCFAs compared with their European counterparts [26].

A meta-analysis reported that dietary fiber interventions increase *Bifidobacterium* and *Lactobacillus* spp. levels without altering α-diversity [27]. High-fiber diets are linked to elevated SCFA production in the gut [28]. Diminished fiber intake not only reduces SCFA production but also prompts gut microbiota to use less optimal substrates, such as amino acids and host mucins, for energy [29]. A diet abundant in plant polysaccharides stimulates the proliferation of *Bacteroidetes* while suppressing *Firmicutes* [26].

The alterations observed in the microbiota due to dietary changes described above, such as high-fat dietary patterns, high intake of MUFA, and a low-fiber diet, lead to disruption of the equilibrium of microbial flora. This disruption promotes the proliferation of bacterial species linked to CRC onset, such as *Proteobacteria*, *Bacteroides* spp. and consequently diminishes the species diversity found in individuals with good health [14].

Yu et al. studied the impact of free amino acid (FAA)-based diets on gut microbiota and cancer progression in BALB/c and ApcMin/+ C57BL/6 mice, using both chemically induced and spontaneous CRC models. They compared the effects of a casein protein diet (CTL) to an FAA-based diet on CRC progression, gut microbiota, and metabolites. The FAA diet significantly reduced CRC progression, as evidenced by less colonic shortening and fewer histopathological changes compared with the CTL diet. It also enriched beneficial gut bacteria such as *Akkermansia* and *Bacteroides*, reversing CRC-associated dysbiosis. Metabolomic analysis showed increased levels of ornithine cycle metabolites and specific FA, such as Docosapentaenoic acid (DPA), in FAA-fed mice. Transcriptomic analysis revealed that FAA upregulated Egl-9 family hypoxia inducible factor 3 (Egln 3) and downregulated several cancer-related pathways, including Hippo, mTOR, and Wnt signaling. Additionally, DPA was found to significantly induce EGLN 3 expression in CRC cell lines. The study suggests that FAA-based diets can modulate gut microbiota, enhance protective metabolites, improve gut barrier functions, and inhibit carcinogenic pathways, thereby decelerating CRC progression [30].

## 5. Dysbiosis

Changes in both composition and metabolic activity within the intestinal microbiota are termed dysbiosis [31]. The gut microbiota undergoes natural fluctuations influenced by variations in nutrient availability, medication use, immune responses, and intestinal mucosal conditions. Minor alterations in microbial balance can create conditions favoring the proliferation of certain bacterial groups, exacerbated by stressors like oxidative stress, bacteriophage induction, and bacteriocin secretion (Figure 3), ultimately leading to decreased microbial diversity and increased Proteobacteria abundance [32,33].

## 6. The Impact of Dysbiosis on CRC Development

Dysbiosis in the gut microbiota is a key driver of inflammation and is strongly associated with CRC development [15]. Approximately 15–20% of all cancer cases are attributed to viral or bacterial infections [34]. Infectious agents, particularly viruses, can promote tumor growth by inducing chronic inflammation, transferring active oncogenes, or promoting immunosuppression [35]. Microbial pathogens may directly contribute to tumorigenesis through DNA-damaging substances, such as nitric oxide or reactive oxygen [32]. For example, *Enterococcus faecalis*, a commensal bacterium living in the colon, produces extracellular superoxide, which can cause chromosomal instability, DNA damage, and malignant transformation [36]. Various hypotheses suggest mechanisms through which bacteria contribute to CRC, including dysbiotic microbial communities driving pro-inflammatory responses and epithelial cell transformation. Another theory, known as the “driver–passenger” model (Figure 4), posits that intestinal bacteria, termed “bacteria drivers”, initiate CRC by inducing epithelial DNA damage and tumorogenesis, thereby facilitating the proliferation of passenger bacteria that thrive in the tumor microenvironment [32,37]. Some bacterial species were identified as potentially playing a role in the carcinogenesis process of CRC, including *Escherichia coli*, *Fusobacterium* spp., *Bacteroides fragilis*, *Streptococcus bovis*, *Enterococcus faecalis*, *Helicobacter pylori*, *Clostridium septicum* [15].

One well-studied bacterium in CRC carcinogenesis is *Fusobacterium nucleatum (F. nucleatum)*. Gur et al. demonstrated that *F. nucleatum* promotes inflammatory responses and inhibits natural killer (NK) cell killing of tumors. The research revealed that this inhibition is mediated by the human TIGIT (T cell immunoglobulin and ITIM domain) receptor, present on NK and various T cells. The Fap2 protein of *F. nucleatum* interacts with TIGIT, reducing NK cell cytotoxicity. Additionally, tumor-infiltrating lymphocytes expressing TIGIT also showed inhibited activity. The study highlighted a mechanism where *F. nucleatum* uses the Fap2 protein to evade the immune system, thereby supporting tumor progression in CRC [38].

Rubinstein et al. discovered that *F. nucleatum* promotes CRC by adhering to, invading, and inducing oncogenic and inflammatory responses in CRC cells via its FadA adhesin. FadA binds to E-cadherin, activating β-catenin signaling and modulating inflammatory and oncogenic pathways. An 11 amino acid region on E-cadherin as the FadA-binding site was identified. A synthetic peptide from this region inhibited FadA-induced CRC cell growth and responses. FadA levels were significantly higher in patients with adenomas and adenocarcinomas compared with normal individuals, correlating with increased expression of oncogenic and inflammatory genes. FadA is thus a potential diagnostic and therapeutic target for CRC [39].

In a further study, Rubinstein et al. investigated how *F. nucleatum* promotes colorectal cancer (CRC) by inducing Annexin A1, a modulator of Wnt/β-catenin signaling. This in vitro cancer progression model showed that *F. nucleatum* stimulates the growth of CRC cells but not precancerous adenoma cells. Annexin A1, specifically expressed in proliferating CRC cells, activates Cyclin D1 and is a predictor of poor prognosis. The study found that F. nucleatum’s FadA adhesin upregulates Annexin A1 via E-cadherin, establishing a positive feedback loop in cancerous cells. This loop is absent in non-cancerous cells, highlighting Annexin A1 as a crucial factor in CRC progression [40].

Another known agent involved in carcinogenesis and progression of CRC is colibactin-producing E. Coli. Shine et al. conducted a study on U2OS and HeLa cell lines. They demonstrated that free colibactins cause DNA double-strand breaks (DSBs) in human cell cultures without needing bacterial delivery. Using domain-targeted editing, they found that some native colibactins, produced through module skipping in the nonribosomal peptide synthetase–polyketide synthase (NRPS–PKS) pathway, exhibit DNA alkylation similar to model colibactins but lack strong DNA interstrand cross-linking. This biosynthetic diversity results in metabolites with different actions (DNA alkylation vs. cross-linking). Membranes between human cells and colibactins reduced genotoxicity, suggesting that membrane diffusion limits colibactin activity and explains the necessity for direct bacterium–human cell contact. Additionally, the colibactin resistance protein ClbS intercepted colibactins in an E. coli–human cell infection model. The study identified protein domains crucial for DNA cross-linking and highlights the diverse mechanisms of colibactins in cell cultures [41].

## 7. The Impact of the Gut Microbiota on the Treatment Toxicity in CRC

The influence of gut microbiota on CRC extends beyond its carcinogenic potential to encompass its role to modulate the efficacy and toxicity of cancer treatments [42].

Kawasaki et al. conducted a prospective observational study involving 23 CRC patients receiving first-line CTH containing Fluoropyrimidines (FPs). Exclusion criteria included recent antibiotic use or initiation/change of oral probiotics prior to CTH. Patients were categorized into diarrhea and non-diarrhea groups based on symptoms. Treatment regimens included TS-1^®^, capecitabine, and 5-FU, combined with OXA or other agents like bevacizumab and panitumumab. TS-1^®^ and capecitabine were administered orally, while 5-FU was given intravenously. In a group of patients receiving oral FPs, the gut microbiome was assessed before and after CTH administration. While no significant changes were observed in alpha and beta diversity post-CTH, the diarrhea group exhibited a notable decrease in *Firmicutes* abundance and a corresponding increase in *Bacteroidetes* abundance at the phylum level. Additionally, the abundance of *Bifidobacterium* significantly decreased in this group. Conversely, in the non-diarrheal group, *Actinobacteria* abundance increased significantly at the phylum level post-chemotherapy, along with significant increases in *Bifidobacterium*, *Fusicatenibacter*, and *Dorea* abundance at the genus level. In a group of four patients who received intravenous FPs, no notable alterations were observed in either alpha or beta diversity following chemotherapy. Furthermore, no significant changes were detected in the microbiome composition at the phylum or genus level. The study indicated a correlation between alterations in the intestinal microbiome and CTH-induced diarrhea, including treatments with FPs. Notably, organic-acid-producing bacteria may play a role in this association [43].

Rice et al. investigated the impact of probiotic supplementation on chronic inflammation in mice with colorectal carcinogenesis undergoing 5FU-based chemotherapy. Mice were divided into five groups: healthy control, colitis model, CTH alone, CTH with probiotics, and probiotics alone. To induce colitis-associated CRC, mice were treated with 1,2-dimethylhydrazine (DMH). Chemotherapy was administered intraperitoneally, while probiotics were given daily via gavage. Probiotic supplementation reduced dysplasia and premalignant lesions, mitigated mucin depletion, and inhibited NFκB expression, suggesting a regulatory role in inflammatory responses. However, probiotics also increased the expression of the cell proliferation marker Ki-67 and caused weight loss and colon shortening [44].

Ziemson et al. examined the impact of capecitabine treatment on fecal SCFA and branched-chain fatty acid (BCFA) levels in CRC patients, along with their associations with various clinical parameters. The study included 44 patients with metastatic or unresectable CRC undergoing capecitabine treatment. Significant reductions in fecal SCFA levels were observed during treatment cycles, while baseline BCFA levels correlated with tumor response. However, nutritional status, physical performance, and chemotherapy-induced toxicity did not demonstrate significant associations with SCFA or BCFA levels. Additionally, baseline SCFA levels correlated positively with blood neutrophil counts, and specific bacterial families showed associations with SCFA/BCFA levels at different time points during treatment. These findings suggest potential links between gut microbiota, SCFA/BCFA levels, and clinical outcomes in CRC patients undergoing capecitabine treatment. During capecitabine treatment, fecal valerate and caproate levels decrease, while acetate, propionate, and butyrate concentrations remain constant. This stability in the latter SCFAs may be attributed to their production by a broader spectrum of gut bacteria. Therefore, if the abundance of dominant SCFA-producing bacteria changes during capecitabine treatment, other bacterial species may potentially assume the role of SCFA production to maintain gut homeostasis. The study underscores the potential significance of SCFA and BCFA in capecitabine treatment, highlighting the need for further research to understand better their roles, as well as facilitate evidence-based interventions targeting gut microbiota and SCFA/BCFA production during CTH in CRC patients [45].

He et al. studied the impact of gut microbiota on CTH toxicity in mouse models. Specific pathogen-free (SPF) C57BL/6 mice exposed to high-dose OXA exhibited increased weight loss (WL) and worsened clinical symptoms, with only half surviving long-term. High-dose CTH resulted in decreased blood cell counts and compromised hematopoietic and gastrointestinal function. Fecal microbiota transplantation (FMT) from OXA-exposed mice exacerbated toxicity in recipient mice. Additionally, OXA-induced toxicity worsened in Il10−/− mice, while macrophage-derived IL-10 alleviated toxicity. Depletion of *Lactobacillus* and *Bifidobacterium* intensified toxicity, whereas probiotic supplementation mitigated adverse effects without compromising CTH efficacy. Investigators collected blood samples before and after the fourth cycle of CTH among CRC patients treated with OXA. Analysis of samples demonstrated reduced levels of peripheral CD45 + IL-10+ cells. These findings highlight the role of microbiota-mediated IL-10 production in CTH tolerance, suggesting its potential as a clinical target [46].

Wang et al. investigated the effects of Xiao-Chai-Hu-Tang (XCHT), a traditional Chinese formula, on alleviating CPT-11-induced toxicity, particularly diarrhea, in FVB mice. The study revealed that CPT-11 administration led to severe diarrhea, characterized by increased disease activity index (DAI) and decreased body weight, but treatment with XCHT significantly alleviated diarrhea symptoms. Analysis showed that XCHT increased the abundance of beneficial bacteria like *Bacteroidetes* and *Firmicutes* while decreasing harmful bacteria like *Proteobacteria*. Specifically, XCHT administration led to a notable rise in the abundance of *Lachnospiraceae_NK4A136_group*, *Lactobacillus*, *Bacteroides*, *Alloprevotella*, and *Prevotellaceae_UCG-001*, which are known for their beneficial effects on gut health. Conversely, harmful bacteria such as *Proteus*, *Parabacteroides*, *Lachnoclostridium*, *Anaerotruncus*, and *Bilophila* showed significant reductions following XCHT treatment. These changes in bacterial composition suggest that XCHT exerts its protective effects by promoting the growth of beneficial bacteria while suppressing the proliferation of harmful ones. These findings highlight the potential of XCHT as a therapeutic strategy for preventing CTH-induced diarrhea by modulating the gut microbiota [47]. The research provided an intriguing perspective on an alternative approach to enhance gut health in CRC patients receiving chemotherapy with CPT-11.

Yue et al. studied the potential of *Bifidobacterium longum (B. longum)* SX-1326 to mitigate gastrointestinal toxicity triggered by CPT-11 in a CRC mouse model. The study involved 50 mice randomly assigned to five groups, each receiving different treatments. These treatments included CRC induction, CPT-11 administration alone, *B. longum* SX-1326 gavage, or a combination of CPT-11 and *B. longum* SX-1326. Alterations in weight were monitored weekly, and fecal samples, blood, colon tissue, and brain tissue were collected for analysis. A kaolin experiment was conducted to evaluate the impact of B. longum SX-1326 on vomiting post-chemotherapy. The results demonstrated reduced kaolin intake in mice treated with *B. longum* SX-1326 post-chemotherapy compared with those receiving chemotherapy alone. Immunohistochemical staining of c-FOS protein in the mouse brain indicated decreased expression in mice receiving the combined treatment. Further examination revealed modulation of neurotransmitter expression, with reduced serotonin (5-HT) and Substance P (SP) levels in the colon of mice treated with *B. longum* SX-1326. Additionally, alterations in inflammatory marker expression and intestinal barrier function were observed. Treatment with *B. longum* SX-1326 led to decreased expression of inflammatory proteins and restoration of mucin expression, reducing intestinal inflammation and improving barrier function. Overall, the findings suggest that *B. longum* SX-1326 holds promise in ameliorating chemotherapy-induced gastrointestinal toxicity in CRC mouse models by modulating neurotransmitter expression, reducing inflammation, and restoring intestinal barrier function [48]. Further investigation in clinical settings is needed.

Mahdy et al. investigated the impact of CPT-11 on gut microbiota composition and the role of probiotics in limiting CPT-11-associated diarrhea and suppressing gut bacterial β-glucuronidase enzymes. Stool samples from 15 volunteers were analyzed, including healthy individuals, colon cancer patients, and individuals being treated with CPT-11. The study revealed significant microbiota perturbations in colon cancer and CPT-11-treated groups compared with healthy individuals. Notably, specific bacterial taxa were more abundant in healthy individuals, including *Actinobacteria*, *Verrucomicrobia*, *Bifidobacterium*, *Gimmiger*, *and Phascolarctobacterium.* In contrast, other bacteria increased in colon cancer and Irinotecan-treated groups, such *as Lactobacillus, Veillonella, Clostridium, Butryicicoccus,* and *Prevotella*. Various probiotic strains and combinations were tested in a mouse model alongside CPT-11 administration. The mice’s colon samples were analyzed for oxidative stress and inflammatory biomarkers. Additionally, clinical isolates of *E. coli* were examined for β-glucuronidase activity, with all three tested strains showing enzymatic activity. Further investigation into the inhibitory effects of probiotics on β-glucuronidase expression revealed that the mixture of *L. plantarum*, *L. acidophilus*, *and L. rhamnosus* exhibited the most significant reduction. Moreover, this combination provided superior antioxidant and anti-inflammatory effects compared with individual strains, as evidenced by histological analysis indicating restoration of normal colon tissue structure with probiotic administration. The study highlights the potential of probiotics to mitigate CPT-11-induced toxicity through multiple mechanisms, including suppression of β-glucuronidase activity, reduction of oxidative stress and inflammation, and preservation of mucosal integrity. Future research should focus on refining probiotic regimens for optimal efficacy and elucidating their precise mechanisms of action in preventing CTH-induced gastrointestinal complications [49]. The impact of individual bacterial strains on the toxicity of cytostatics in the discussed studies is presented in Table 1.

These insights highlight the potential for personalized approaches to CRC management, integrating microbiota modulation strategies to optimize treatment efficacy and minimize adverse effects. 

## 8. The Role of Probiotics in CRC Therapy

According to the Food and Agriculture Organization of the United Nations and the World Health Organization (WHO) (FAO/WHO), probiotics are “live microorganisms that, when administered in adequate amounts, confer a health benefit on the host” [50].

There are available data regarding the efficacy and relevance of adjunctive probiotic use in the treatment of CRC patients [51,52,53].

Huang et al. conducted a study involving locally advanced 100 CRC patients aged 40–70 and without severe comorbidities, scheduled for surgical resection followed by CTH. Exclusion criteria included recent antibiotic or probiotic use, metabolic diseases, severe comorbidities, history of inflammatory bowel disease or colorectal adenoma, family history of CRC or gastrointestinal cancers, hypersensitivity to study drugs, or pregnancy. Patients were randomly allocated into two groups post-surgery. The Probio group received oral probiotic tablets (with Combined *B. infants*, *L. acidophilus*, *E. faecalis*, and *B. cereus*), while the Placebo group received placebo tablets for approximately six weeks, including two weeks of CTH. Gastrointestinal symptoms and infections within six weeks were recorded. Blood samples were collected for routine analysis. Fecal samples were obtained at various time points for further analysis. No infections requiring antibiotics were reported during the intervention period. Interestingly, the Probio group experienced markedly fewer episodes of abdominal pain, distention, constipation, and diarrhea compared with the Placebo group. However, the two groups had no significant differences in CTH efficacy. The Placebo group exhibited higher levels of *Akkermansia* and *Lachnospiraceae_Clostridium* but lower levels of *Prevotella*, *Lactobacillus*, and *Roseburia* compared with the Probio group. Interestingly, probiotic supplementation restored the abundance of these genera closer to healthy levels and promoted beneficial genera like *Bifidobacterium*, *Streptococcus*, and *Blautia* while reducing potentially harmful ones such as *Faecalibacterium*, *Fusobacterium*, *Sutterella*, and *Megamonas*. Probiotic administration reversed changes in gut microbiota induced by surgery and CTH, particularly in the abundance of specific bacterial genera. Probiotics also increased SCFA levels in fecal samples. Correlation analysis revealed associations between specific bacterial genera and SCFA levels. Ultimately, probiotic supplementation effectively mitigated CTH-induced gastrointestinal complications and modulated gut microbiota composition in CRC patients [54].

Bajramagic et al. investigated the impact of probiotic therapy on postoperative complications among 78 CRC patients undergoing surgical resection. The treatment group received oral probiotics for one-year after surgery, containing eight bacterial cultures (*Lactobacillus acidophilus*, *Lactobacillus casei*, *Lactobacillus plantarum*, *Lactobacillus rhamnosus*, *Bifidobacterium lactis*, *Bifidobacterium bifidum*, *Bifidobacterium breve*, *Streptococcusthermophilus*). Complications, particularly ileus, were significantly reduced in the probiotic group compared with the control group without probiotics. Additionally, surgical site infections (SSI) had a higher probability of occurrence in the non-probiotic group, although without significance. Meanwhile, Individuals with probiotic treatment experienced fewer deaths and shorter hospital stays postoperatively. Notably, probiotics were most effective in reducing complications in rectal tumors, suggesting their potential benefit in specific tumor localization [55].

Kotzampassi et al. conducted a randomized controlled trial investigating the effects of a probiotic preparation on postoperative complications following open elective colonic surgery for cancer. In total, 164 patients were randomized to receive either a probiotic formulation containing four strains (*Lactobacillus acidophilus*, *L. plantarum*, *Bifidobacterium lactis* and *Saccharomyces boulardii*) (group of 80 patients) or a placebo (group of 84 patients), starting one day before surgery and continuing for 15 days afterward. The primary outcome was the development of postoperative complications within 30 days. The study was terminated prematurely due to the probiotics’ effectiveness, significantly reducing the incidence of major complications compared with the placebo. Specifically, probiotics notably decreased rates of pneumonia, SSIs, and AL. The probiotics group also experienced a shorter hospital stay. Gene expression analysis revealed a correlation between the probiotics and the regulation of TNF and IL-6. Overall, the probiotics significantly mitigated the risk of postoperative complications, suggesting modulation of gene expression, particularly of SOCS3, as a potential underlying mechanism [56].

Carlini et al. conducted a pilot study on a group of 60 patients who underwent elective colorectal laparoscopic surgery for tumors (both benign and malignant) and introduced the Microbiota Implementation to Reduce Anastomotic Colorectal Leaks (MIRACLe) protocol (Figure 5), which was comprised of several key components: oral antibiotic prophylaxis (instead of intravenous), low-volume mechanical bowel preparation (MBP), and perioperative administration of probiotics. The control group consisted of 500 patients treated with a standard Enhanced Recovery After Surgery (ERAS) protocol. In the preoperative phase, the MIRACLe group of patients were given oral antimicrobial prophylaxis, comprising amoxicillin-clavulanic acid at 1 g every 12 h and metronidazole at 250 milligrams every 8 h. In the case of penicillin allergy, oral ciprofloxacin was administered at 500 milligrams twice daily. Patients underwent mechanical bowel preparation using a polyethylene glycol solution. Additionally, oral probiotics were administered starting five days before surgery, consisting of a combination of eight different strains of bacteria (*Streptococcus thermophilus*, *Bifidobacterium breve*, *Bifdobacterium longum*, *Bifdobacterium infantis*, *Lactobacillus acidophilus*, *Lactobacillus plantarum*, *Lactobacillus paracasei*, *Lactobacillus delbrueckii subsp. Bulgaricus*). During the intraoperative phase, patients continued with oral antibiotic prophylaxis, and two doses of probiotics dissolved in a sterile physiological solution were instilled into the lumen of the proximal anastomotic bowel stump. In the postoperative phase, patients received continued oral probiotics for up to four days after surgery. The protocol’s implementation resulted in improvements in clinical outcomes compared with the control group, including reduced incidence of AL (1.7% vs. 6.4%; study group vs. control group), SSIs (1.7% vs. 3.6%), and reoperation rates (1.7% vs. 4.2%), as well as shorter recovery times and hospital stays. However, the variance did not attain significance due to the limited number of cases (60 vs. 500 patients in the study group and control group, respectively). Overall, the MIRACLe protocol demonstrated safety and efficacy in enhancing patient outcomes following laparoscopic colorectal resection for cancer. Its comprehensive approach, addressing various stages of the perioperative period, proved instrumental in reducing surgical complications and improving postoperative recovery, highlighting its potential as a valuable clinical intervention in colorectal surgery [57].

Marcellinaro et al. aimed to ascertain the safety and effectiveness of the MIRACLe protocol [58]. The study encompassed two groups: a prospective group of 131 patients, including the initial 60 cases [57], and a retrospective group of 500 patients who underwent standard ERAS preparation. To mitigate discrepancies, propensity score matching was conducted, resulting in post-matched groups of 118 patients in the MIRACLe group and 354 patients in the control group. The primary objective was to assess AL incidence, with secondary outcomes encompassing SSIs, surgical complications, readmission, reoperation rates, clinical outcomes, mean hospital stay, and postoperative mortality. Patients in the control group received intravenous antibiotic prophylaxis and mechanical bowel preparation, while individuals in the MIRACLe group followed the MIRACLe protocol. Surgical procedures were standardized across both groups. Postoperatively, both adhered to ERAS protocol guidelines. The MIRACLe group exhibited significantly lower drainage placement post-surgery, with no notable differences in conversion to open surgery. Adverse reactions related to intraoperative probiotic administration were absent in the MIRACLe patients. Post-matching, the MIRACLe group demonstrated reduced surgical complications, notably AL rates. Faster recovery times for bowel movements, oral feeding, and shorter hospital stays were observed in the MIRACLe group. The protocol proved safe and effective [58].

## 9. Microbiome Gene Expression and Its Relevance in CRC

With the advancement of new technologies, new avenues are opening for medical investigation, including the field of CRC research. Rohani et al. conducted a comprehensive study investigating the interplay between *Escherichia coli* K-12, CRC, and gene expression alterations. Initially, bioinformatics tools were utilized to identify genes influenced by *E. coli* K-12. Subsequently, the relationship between these genes and CRC-related genes was explored using The Cancer Genome Atlas (TCGA) data. Primary targets, selected as hub genes associated with both *E. coli* K-12 and CRC, underwent expression analysis via Reverse Transcription Quantitative Polymerase Chain Reaction (RT-qPCR) in CRC samples compared with adjacent normal tissues. For deeper insights, CRC cell lines were cultured with *E. coli* strains, and gene expression profiles were analyzed. Differential expression analyses and pathway enrichment studies revealed significant alterations in inflammation, hypoxia, apoptosis, and DNA repair pathways. Moreover, candidate genes were identified and correlated with patient survival. Notably, *E. coli* K-12 modulated the expression of genes associated with oncogenic and tumor-suppressive roles. Protein–protein interaction network analysis pinpointed key genes, including *BGN*, *FBLN1*, *CXCL3*, and *KLF4*, as potential hubs in the CRC-*E. coli* K-12 axis. Experimental validation through RT-qPCR and immunohistochemistry confirmed the expression changes of select genes in CRC samples compared with normal tissues and their modulation by *E. coli* K-12. Overall, the study suggests a potential inhibitory role of *E. coli* K-12 in CRC development by modulating the expression of oncogenes and tumor suppressors, paving the way for further investigation into its therapeutic potential [59].

## 10. Conclusions

This review highlights the relationship between gut microbiota and CRC development, emphasizing their roles in pathogenesis and treatment. Gut microbiota composition, dietary patterns, dysbiosis, and microbial regulation significantly influence CRC progression and therapy. Dysbiosis contributes to inflammation and CRC advancement, while diet impacts gut microbiota composition. Recent research reveals the bidirectional effects between gut microbiota and chemotherapy toxicity, suggesting personalized treatment strategies with probiotics. Additionally, the MIRACLe protocol in colorectal surgery shows promise for improving surgical outcomes. Further investigation into personalized probiotic supplementation in CRC therapy is needed.

## Figures and Tables

**Figure 1 cancers-16-02236-f001:**
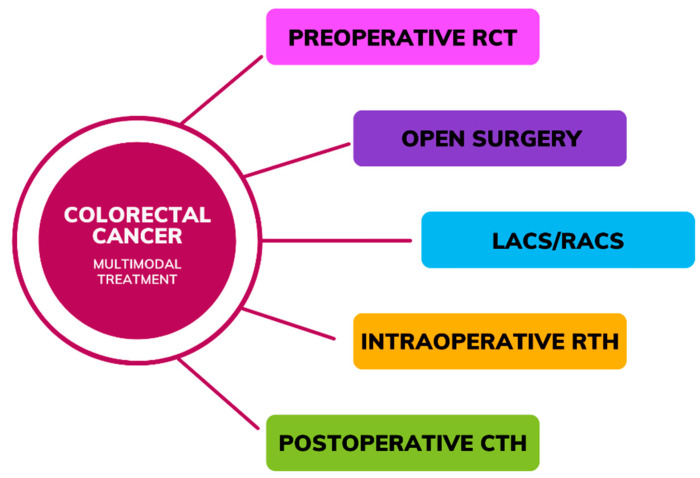
Multimodal treatment in colorectal cancer (RCT—Radio-chemotherapy, LACS—laparoscopic-assisted colorectal surgery, RACS—robot-assisted colorectal surgery, RTH—Radiotherapy, CTH—Chemotherapy).

**Figure 2 cancers-16-02236-f002:**
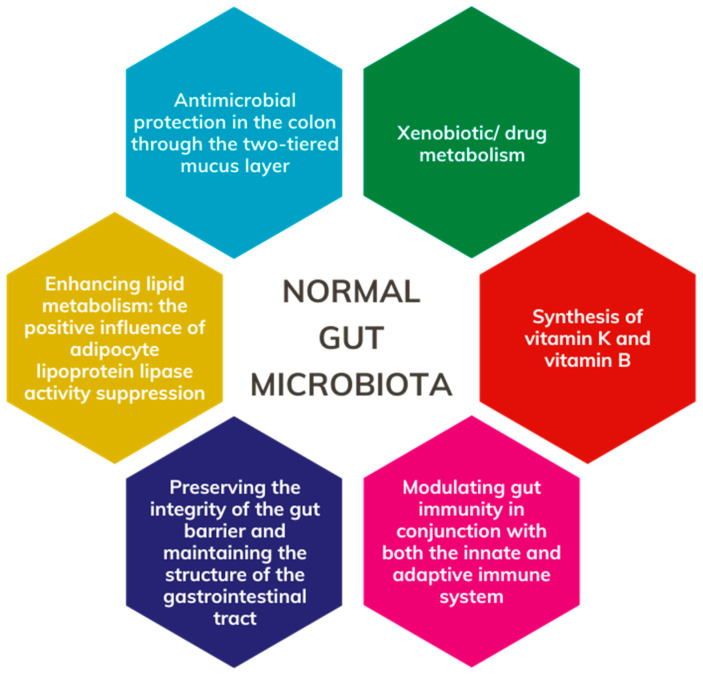
Role of the normal gut microbiota.

**Figure 3 cancers-16-02236-f003:**
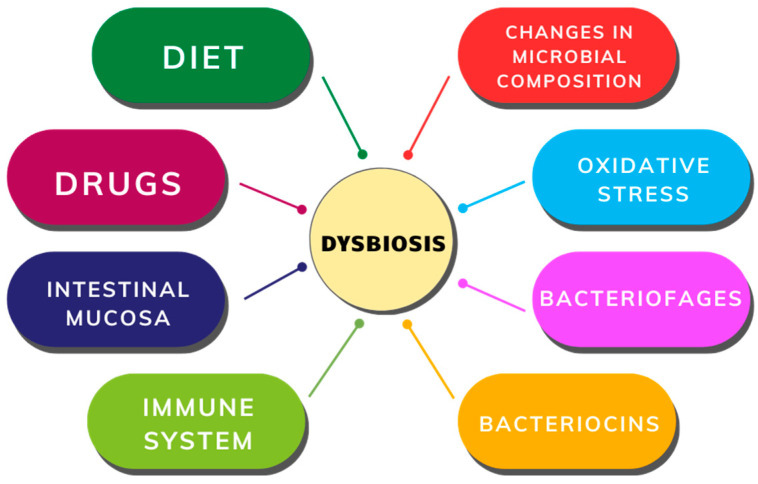
Factors contributing to gut dysbiosis.

**Figure 4 cancers-16-02236-f004:**
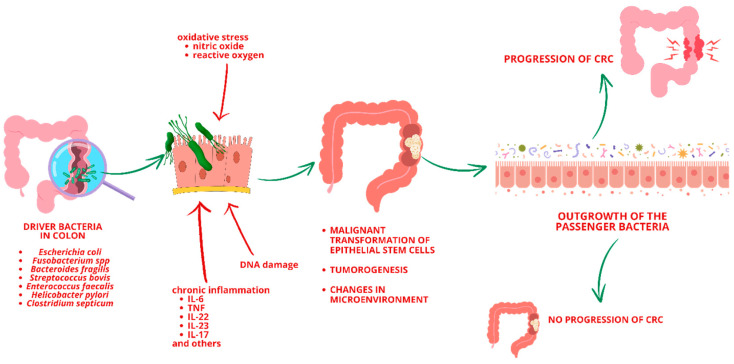
“Driver–Passenger” model (CRC—Colorectal Cancer).

**Figure 5 cancers-16-02236-f005:**
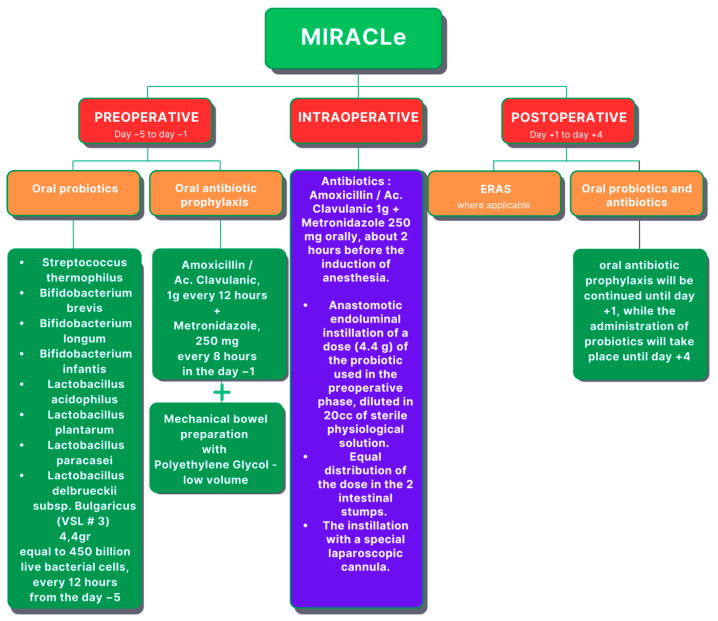
The microbiota implementation to reduce anastomotic colorectal leaks (MIRACLe) protocol (ERAS—Enhanced Recovery After Surgery).

**Table 1 cancers-16-02236-t001:** The impact of the gut microbiota on the toxicity of cytostatics in colorectal cancer.

Bacteria Strain	Abundance of Bacteria	Cytostatic	Impact on Toxicity of Treatment
*Bacteroidetes* [43]	↑	Fluoropyrimidines	intensified diarrhea
*Firmicutes*, *Bifidobacterium* [43]	↓	Fluoropyrimidines	intensified diarrhea
*Actinobacteria*, *Bifidobacterium*, *Fusicatenibacter*, *Dorea* [43]	↑	Fluoropyrimidines	decreased diarrhea
*Lactobacillus* and *Bifidobacterium* [46]	↓	Oxaliplatin	intensified toxicity
*Proteus*, *Parabacteroides*, *Lachnoclostridium*, *Anaerotruncus*, and *Bilophila* [47]	↑	Irinotecan	intensified diarrhea
*Lachnospiraceae_NK4A136_group, Lactobacillus*, *Bacteroides*, *Alloprevotella*, *Prevotellaceae_UCG-001*, *Actinobacteria*, *Verrucomicrobia*, *Bifidobacterium*, *Gimmiger*, and *Phascolarctobacterium* [48]	↓	Irinotecan	intensified diarrhea
*Bifidobacterium longum* SX-1326 [48]	↑	Irinotecan	decreased vomiting
*L. plantarum*, *L. acidophilus*, and *L. rhamnosus* [49]	↑	Irinotecan	decreased toxicity

## Data Availability

The data presented in this study are available on request from the corresponding author (accurately indicate status).

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
