# Peer review of "Gut Microbiota—Adversary or Ally? Its Role and Significance in Colorectal Cancer Pathogenesis, Progression, and Treatment"

_cancers, 2024, doi:10.3390/cancers16122236_

Round 1

Reviewer 1 Report

Comments and Suggestions for Authors

Thank you for the opportunity to review this paper

This is a very  interesting review entitled: Gut Microbiota- Adversary or Ally? Its Role and Significance in Colorectal Cancer Pathogenesis, Progression, and Treatment

 There are some topics in the article that should be considered:

Introduction

I consider that the introduction is a bit long and some figures, for example number two, could be eliminated. This topic is dealt with extensively in the results.  

Methodology

Please include more details about search strategy, inclusion and exclusion criteria, quality assesment and data extraction

Results

 In the paragraph that mentions the effect of nutrition on the microbiome, it would be important to refer to studies that have investigated this correlation in colorectal cancer.

In the paragraph concerning the impact of dysbiosis on CRC development It would be valuable to mention some clinical studies that contribute to this theory. The style of description of clinical studies should be the same in all chapters of the article.

I consider it unnecessary at the end of each study described to mention that it is necessary to carry out more clinical studies in this regard; it could be mentioned only once at the end.

The results are presented in a confusing manner, referring to different clinical studies independent of each other; they should be grouped by effects or by microorganisms.

Table 1 refers to which study?Please add the respective references

In the paragraph concerning The role of probiotics in CRC therapy I consider that there should be a small reference to the definition of probiotics

I think that the paragraph that refers to New direction- probiotics in CRC prevention does not correspond to the title, something like the effect of probiotics in microbiome gene expression and its relevance in colorectal cancer would be better. 

This review is interesting but needs to be better organized to be published.

Please check the text, there are several spelling errors

Author Response

23rd May 2024, Lublin

Prof. Dr. Samuel C. Mok

Editor-in-Chief

Cancers

RE: cancers-2999615

Gut Microbiota- Adversary or Ally? Its Role and Significance in Colorectal Cancer Pathogenesis, Progression, and Treatment.

Dear Prof. Dr. Samuel C. Mok,

Thank you for reviewing our Original Article # cancers-2999615 “Gut Microbiota- Adversary or Ally? Its Role and Significance in Colorectal Cancer Pathogenesis, Progression, and Treatment”

We are grateful for valuable insight and comments as these serve to further strengthen our manuscript. As requested, we have provided a point-by-point response to each of the comments with relevant changes made to the manuscript.

 Comments to Author:

Thank you for the opportunity to review this paper

This is a very interesting review entitled: Gut Microbiota- Adversary or Ally? Its Role and Significance in Colorectal Cancer Pathogenesis, Progression, and Treatment

There are some topics in the article that should be considered:

1. Introduction I consider that the introduction is a bit long and some figures, for example number two, could be eliminated. This topic is dealt with extensively in the results
2. Methodology, Please include more details about search strategy, inclusion and exclusion criteria, quality assesment and data extraction
3. Results In the paragraph that mentions the effect of nutrition on the microbiome, it would be important to refer to studies that have investigated this correlation in colorectal cancer
4. In the paragraph concerning the impact of dysbiosis on CRC development It would be valuable to mention some clinical studies that contribute to this theory. The style of description of clinical studies should be the same in all chapters of the article.
5. I consider it unnecessary at the end of each study described to mention that it is necessary to carry out more clinical studies in this regard; it could be mentioned only once at the end.

  1. The results are presented in a confusing manner, referring to different clinical studies independent of each other; they should be grouped by effects or by microorganisms.
  2. Table 1 refers to which study? Please add the respective references.
  3. In the paragraph concerning the role of probiotics in CRC therapy I consider that there should be a small reference to the definition of probiotics.
  4. I think that the paragraph that refers to new direction- probiotics in CRC prevention does not correspond to the title, something like the effect of probiotics in microbiome gene

This review is interesting but needs to be better organized to be published.

Please check the text, there are several spelling errors

Thank you very much for this opportunity.

Dear Reviewer, thank you for this insightful feedback. We appreciate your acknowledgment of the comprehensive overview of the background and rationale for our study.

  1. The authors would like to thank the reviewer for pointing out this important aspect. To emphasize this aspect, the Figure and lines 129-131 were deleted from the Introduction.
  2. Appropriate sentence to clarify this topic was added to Methodology paragraph (lines 140-145).

“In order to maintain quality standards, the included studies underwent evaluation concerning their relevance, methodology, and significance. Systematic data extraction was carried out, encompassing key discoveries, study design specifics, participant characteristics (limited to CRC patients across all stages of the disease, various age groups, or animal subjects), interventions applied, resultant outcomes, and whether the study successfully reached any defined endpoints.”

  1. We would like to thank the Reviewer for this important remark. Appropriate paragraph was added to Results section (lines 207-226).

“The alterations observed in the microbiota due to dietary changes described above, such as high-fat dietary patterns, high intake of MUFA, and a low-fiber diet, lead to disruption of the equilibrium of microbial flora. This disruption promotes the proliferation of bac-terial species linked to CRC onset, such as Proteobacteria, Bacteroides spp. and consequently diminishes the species diversity found in individuals with good health. [14] Yu et al. studied the impact of free amino acid (FAA)-based diets on gut microbiota and cancer progression in BALB/c and ApcMin/+ C57BL/6 mice, using both chemically induced and spontaneous CRC models. They compared the effects of a casein protein diet (CTL) to an FAA-based diet on CRC progression, gut microbiota, and metabolites. The FAA diet significantly reduced CRC progression, as evidenced by less colonic shortening and fewer histopathological changes compared to the CTL diet. It also enriched beneficial gut bacteria such as Akkermansia and Bacteroides, reversing CRC-associated dysbiosis. Metabolomic analysis showed increased levels of ornithine cycle metabolites and specific FA, such as Docosapentaenoic acid (DPA), in FAA-fed mice. Transcriptomic analysis revealed that FAA upregulated Egl-9 family hypoxia inducible factor 3 (Egln 3) and downregulated several cancer-related pathways, including Hippo, mTOR, and Wnt signaling. Additionally, DPA was found to significantly induce EGLN 3 expression in CRC cell lines. The study suggests that FAA-based diets can modulate gut microbiota, enhance protective metabolites, improve gut barrier functions, and inhibit carcinogenic pathways, thereby decelerating CRC progression. [30]”

  1. The mentioned paragraph had been adjusted and the style had been unified. (lines 258-298)

“One well-studied bacterium in CRC carcinogenesis is Fusobacterium nucleatum (F. nucleatum. Gur et al. demonstrated that F. nucleatum promotes inflammatory responses and inhibits natural killer (NK) cell killing of tumors. The research revealed that this inhibition is mediated by the human TIGIT ((T cell immunoglobulin and ITIM domain) receptor, present on NK and various T cells. The Fap2 protein of F. nucleatum interacts with TIGIT, reducing NK cell cytotoxicity. Additionally, tumor-infiltrating lymphocytes expressing TIGIT also showed inhibited activity. The study highlighted a mechanism where F. nucleatum uses the Fap2 protein to evade the immune system, thereby supporting tumor progression in CRC. [38]

Rubinstein et al. discovered that F. nucleatum promotes CRC by adhering to, invading, and inducing oncogenic and inflammatory responses in CRC cells via its FadA adhesin. FadA binds to E-cadherin, activating β-catenin signaling and modulating inflammatory and oncogenic pathways. An 11 amino acid region on E-cadherin as the FadA-binding site was identified. A synthetic peptide from this region inhibited FadA-induced CRC cell growth and responses. FadA levels were significantly higher in patients with adenomas and adenocarcinomas compared to normal individuals, correlating with increased expression of oncogenic and inflammatory genes. FadA is thus a potential diagnostic and therapeutic target for CRC. [39]

In further study Rubinstein et al. investigated how F. nucleatum promotes colorectal cancer (CRC) by inducing Annexin A1, a modulator of Wnt/β-catenin signaling. This in vitro cancer progression model showed that F. nucleatum stimulates the growth of CRC cells but not precancerous adenoma cells. Annexin A1, specifically expressed in proliferating CRC cells, activates Cyclin D1 and is a predictor of poor prognosis. The study found that F. nucleatum's FadA adhesin upregulates Annexin A1 via E-cadherin, establishing a positive feedback loop in cancerous cells. This loop is absent in non-cancerous cells, highlighting Annexin A1 as a crucial factor in CRC progression. [40]

Another known agent involved in carcinogenesis and progression of CRC is colibac-tin-producing E. Coli. Shine et al. conducted a study on U2OS and HeLa cell lines. They demonstrated that free colibactins cause DNA double-strand breaks (DSBs) in human cell cultures without needing bacterial delivery. Using domain-targeted editing, they found that some native colibactins, produced through module skipping in the nonribosomal peptide synthetase–polyketide synthase (NRPS–PKS) pathway, exhibit DNA alkylation similar to model colibactins but lack strong DNA interstrand cross-linking. This biosynthetic diversity results in metabolites with different actions (DNA alkylation vs. cross-linking). Membranes between human cells and colibactins reduced genotoxicity, suggesting that membrane diffusion limits colibactin activity and explains the necessity for direct bacterium-human cell contact. Additionally, the colibactin resistance protein ClbS intercepted colibactins in an E. coli-human cell infection model. The study identified protein domains crucial for DNA cross-linking and highlights the diverse mechanisms of colibactins in cell cultures. [41]”

  1. Appropriate sentences to clarify this topic were deleted according to the Reviewers suggestion.
  2. The suggested amendments were introduced.
  3. The appropriate references were added to the Table 1.
  4. The definition was added to the paragraph concerning the role of probiotics in CRC therapy. (lines 476-478)

“According to the Food and Agriculture Organization of the United Nations and the World Health Organization (WHO) (FAO/WHO) probiotics are “live microorganisms that, when administered in adequate amounts, confer a health benefit on the host”. [50]”

  1. The title of the mentioned paragraph was changed to: ““Microbiome gene expression and its relevance in CRC.”

We trust that modifications contribute to the overall strength of our study.

Thank you for your time and consideration in reviewing our manuscript.

We appreciate the valuable feedback and we have carefully addressed all comments. We believe that the revisions substantially enhance the quality the manuscript. We look forward to the opportunity to contribute to Cancers and are grateful for your guidance in this process.

Sincerely,

Katarzyna Mielniczek

Reviewer 2 Report

Comments and Suggestions for Authors

1. In the introduction section, lines 57 -61, the authors need to include the most recent statistics about colorectal cancer.

2. In the introduction, the authors should provide examples of microbes that are found in normal individuals and compare it with dysbiosed gut microbiome.

3. There are many review articles on colorectal cancer and gut microbiome, please describe the novelty of this article and what is the gap in the literature that authors are trying to fill.

4. The section “4. Influence of the diet on the gut microbiome” should be revised and authors should include the effect of the diet on the gut microbiome and its co-relation with colorectal cancer.

5. As suggested by the title of the review, authors should provide the role of gut microbiome on colorectal cancer pathogenesis and progression. Authors need to include the detailed mechanism of cancer progression mediated through the gut microbiome.

6. The conclusion should be revised.

Author Response

23rd May 2024, Lublin

Prof. Dr. Samuel C. Mok

Editor-in-Chief

Cancers

RE: cancers-2999615

Gut Microbiota- Adversary or Ally? Its Role and Significance in Colorectal Cancer Pathogenesis, Progression, and Treatment.

Dear Prof. Dr. Samuel C. Mok,

Thank you for reviewing our Original Article # cancers-2999615 “Gut Microbiota- Adversary or Ally? Its Role and Significance in Colorectal Cancer Pathogenesis, Progression, and Treatment”

We are grateful for valuable insight and comments as these serve to further strengthen our manuscript. As requested, we have provided a point-by-point response to each of the comments with relevant changes made to the manuscript.

Comments to Author:

1. In the introduction section, lines 57 -61, the authors need to include the most recent statistics about colorectal cancer.
2. In the introduction, the authors should provide examples of microbes that are found in normal individuals and compare it with dysbiosed gut microbiome.
3. There are many review articles on colorectal cancer and gut microbiome, please describe the novelty of this article and what is the gap in the literature that authors are trying to fill.
4. The section “4. Influence of the diet on the gut microbiome” should be revised and authors should include the effect of the diet on the gut microbiome and its correlation with colorectal cancer.
5. As suggested by the title of the review, authors should provide the role of gut microbiome on colorectal cancer pathogenesis and progression. Authors need to include the detailed mechanism of cancer progression mediated through the gut microbiome.

  1. The conclusion should be revised.

Dear Reviewer, thank you for this valuable observation and careful insight into our work.

  1. The data in Introduction has been changed to the most recent available with the new reference. (lines 57-61)

“In 2022, colorectal cancer (CRC) was the third most prevalent malignancy worldwide, with an estimated incidence of 1 926 425cases, representing almost 10% of all cancer diagnoses. Additionally, CRC accounted for 904 019deaths globally, which was the second most common cancer-related reason for mortality thereby constituting 9.3% of all cancer-related deaths. [1]”

  1. 2. The Authors provided the examples of microbiomes suggested by the Reviewer. (lines 102-106 and 111-116)

“Gut microbiota in healthy adults consists mainly of the following anaerobe bacteria species: Bacteroides, Eubacterium, Bifidobacterium, Fusobacterium, Peptostreptococcus, and Atopobium, with less coexistence in the population of aerobic bacteria, such as: Enterococci, Lactobacilli, Enterobacteriaceae, and Streptococci. [12]”

“The following changes in intestinal microbime were found in CRC patients: increased diversity of the Clostridium leptum and C. coccoides subgroups, Bacteroides/Prevotella, Fusobacterium, presence of such pathogenes as Pseudomonas, Helicobacter and Acinetobacter, decreased Bifidobacterium, Faecalibacterium, Blautia butyrate-producing bacteria and Firmicutes and Lactobacillus.[14]”

  1. The following paragraph clarifying this topic has been added to the Introduction. (lines 125-128)

“We aimed to offer a novel perspective on the utilization of gut microbiota in a multimodal treatment approach for CRC, focusing primarily on alleviating post-CTH symptoms and minimizing the risks of intraoperative and postoperative complications.”

  1. The section “Influence of the diet on the gut microbiome” has been adjusted according to the Reviewers suggestion. (lines 207-226)

“The alterations observed in the microbiota due to dietary changes described above, such as high-fat dietary patterns, high intake of MUFA, and a low-fiber diet, lead to disruption of the equilibrium of microbial flora. This disruption promotes the proliferation of bac-terial species linked to CRC onset, such as Proteobacteria, Bacteroides spp. and consequently diminishes the species diversity found in individuals with good health. [14] Yu et al. studied the impact of free amino acid (FAA)-based diets on gut microbiota and cancer progression in BALB/c and ApcMin/+ C57BL/6 mice, using both chemically induced and spontaneous CRC models. They compared the effects of a casein protein diet (CTL) to an FAA-based diet on CRC progression, gut microbiota, and metabolites. The FAA diet significantly reduced CRC progression, as evidenced by less colonic shortening and fewer histopathological changes compared to the CTL diet. It also enriched beneficial gut bacteria such as Akkermansia and Bacteroides, reversing CRC-associated dysbiosis. Metabolomic analysis showed increased levels of ornithine cycle metabolites and specific FA, such as Docosapentaenoic acid (DPA), in FAA-fed mice. Transcriptomic analysis revealed that FAA upregulated Egl-9 family hypoxia inducible factor 3 (Egln 3) and downregulated several cancer-related pathways, including Hippo, mTOR, and Wnt signaling. Additionally, DPA was found to significantly induce EGLN 3 expression in CRC cell lines. The study suggests that FAA-based diets can modulate gut microbiota, enhance protective metabolites, improve gut barrier functions, and inhibit carcinogenic pathways, thereby decelerating CRC progression. [30]”

  1. Section 6. “The impact of dysbiosis on CRC development” was extended as suggested by the Reviewer. (lines 258-297)

“One well-studied bacterium in CRC carcinogenesis is Fusobacterium nucleatum (F. nucleatum. Gur et al. demonstrated that F. nucleatum promotes inflammatory responses and inhibits natural killer (NK) cell killing of tumors. The research revealed that this inhibition is mediated by the human TIGIT ((T cell immunoglobulin and ITIM domain) receptor, present on NK and various T cells. The Fap2 protein of F. nucleatum interacts with TIGIT, reducing NK cell cytotoxicity. Additionally, tumor-infiltrating lymphocytes expressing TIGIT also showed inhibited activity. The study highlighted a mechanism where F. nucleatum uses the Fap2 protein to evade the immune system, thereby supporting tumor progression in CRC. [38]

Rubinstein et al. discovered that F. nucleatum promotes CRC by adhering to, invading, and inducing oncogenic and inflammatory responses in CRC cells via its FadA adhesin. FadA binds to E-cadherin, activating β-catenin signaling and modulating inflammatory and oncogenic pathways. An 11 amino acid region on E-cadherin as the FadA-binding site was identified. A synthetic peptide from this region inhibited FadA-induced CRC cell growth and responses. FadA levels were significantly higher in patients with adenomas and adenocarcinomas compared to normal individuals, correlating with increased expression of oncogenic and inflammatory genes. FadA is thus a potential diagnostic and therapeutic target for CRC. [39]

In further study Rubinstein et al. investigated how F. nucleatum promotes colorectal cancer (CRC) by inducing Annexin A1, a modulator of Wnt/β-catenin signaling. This in vitro cancer progression model showed that F. nucleatum stimulates the growth of CRC cells but not precancerous adenoma cells. Annexin A1, specifically expressed in proliferating CRC cells, activates Cyclin D1 and is a predictor of poor prognosis. The study found that F. nucleatum's FadA adhesin upregulates Annexin A1 via E-cadherin, establishing a positive feedback loop in cancerous cells. This loop is absent in non-cancerous cells, highlighting Annexin A1 as a crucial factor in CRC progression. [40]

Another known agent involved in carcinogenesis and progression of CRC is colibac-tin-producing E. Coli. Shine et al. conducted a study on U2OS and HeLa cell lines. They demonstrated that free colibactins cause DNA double-strand breaks (DSBs) in human cell cultures without needing bacterial delivery. Using domain-targeted editing, they found that some native colibactins, produced through module skipping in the nonribosomal peptide synthetase–polyketide synthase (NRPS–PKS) pathway, exhibit DNA alkylation similar to model colibactins but lack strong DNA interstrand cross-linking. This biosynthetic diversity results in metabolites with different actions (DNA alkylation vs. cross-linking). Membranes between human cells and colibactins reduced genotoxicity, suggesting that membrane diffusion limits colibactin activity and explains the necessity for direct bacterium-human cell contact. Additionally, the coli-bactin resistance protein ClbS intercepted colibactins in an E. coli-human cell infection model. The study identified protein domains crucial for DNA cross-linking and highlights the diverse mechanisms of colibactins in cell cultures. [41]”

  1. The conclusion has been revised. (lines 617-625)

“This review highlights the relationship between gut microbiota and CRC development, emphasizing their roles in pathogenesis and treatment. Gut microbiota composition, dietary patterns, dysbiosis, and microbial regulation significantly influence CRC progression and therapy. Dysbiosis contributes to inflammation and CRC advancement, while diet impacts gut microbiota composition. Recent research reveals the bidirectional effects between gut microbiota and chemotherapy toxicity, suggesting personalized treatment strategies with probiotics. Additionally, the MIRACLe protocol in colorectal surgery shows promise for improving surgical outcomes. Further investigation into personalized probiotic supplementation in CRC therapy is needed.”

Your feedback has been invaluable in refining this aspect of our manuscript, and we appreciate your guidance.

Thank you for your time and consideration in reviewing our manuscript.

We appreciate the valuable feedback and we have carefully addressed all comments. We believe that the revisions substantially enhance the quality the manuscript. We look forward to the opportunity to contribute to Cancers and are grateful for your guidance in this process.

Sincerely,

Katarzyna Mielniczek

Round 2

Reviewer 1 Report

Comments and Suggestions for Authors

The authors have made most of the suggested corrections. I consider that the article can be accepted for publication.